# Improving HIV pre-exposure prophylaxis (PrEP) adherence and retention in care: Process evaluation and recommendation development from a nationally implemented PrEP programme

Jennifer MacDonald[1], Claudia S. Estcourt[1,2]*, Paul Flowers[3], Rak Nandwani[2,4], Jamie Frankis[1], Ingrid Young[5], Dan Clutterbuck[6], Jenny Dalrymple[1], Lisa McDaid[7], Nicola Steedman[8], John Saunders[9]

1 Research Centre for Health (ReaCH), Glasgow Caledonian University, Glasgow, Scotland, United Kingdom, 2 Sandyford Sexual Health Services, NHS Greater Glasgow and Clyde, Glasgow, Scotland, United Kingdom, 3 School of Psychological Sciences and Health, University of Strathclyde, Glasgow, Scotland, United Kingdom, 4 College of Medical, Veterinary & Life Sciences, University of Glasgow, Glasgow, United Kingdom, 5 Centre for Biomedicine, Self & Society, University of Edinburgh, Edinburgh, Scotland, United Kingdom, 6 Chalmers Sexual Health Centre, NHS Lothian, Edinburgh, Scotland, United Kingdom, 7 Institute for Social Science Research, The University of Queensland, Brisbane, Australia, 8 Chief Medical Officer Directorate, Scottish Government, Edinburgh, Scotland, United Kingdom, 9 Institute for Global Health, University College London, London, England, United Kingdom

* claudia.estcourt@gcu.ac.uk

**Data Availability Statement:** Due to the sensitive nature of the questions asked in this study,

## Abstract

### Introduction

HIV pre-exposure prophylaxis (PrEP), in which people take HIV medication to prevent HIV acquisition, underpins global HIV transmission elimination strategies. Effective prevention needs people to adhere to PrEP and remain in care during periods of risk, but this is difficult to achieve. We undertook a process evaluation of Scotland's PrEP programme to explore barriers and facilitators to PrEP adherence and retention in care and to systematically develop evidence-based, theoretically-informed recommendations to address them.

### Methods

We conducted semi-structured interviews and focus groups (09/2018-07/2019) with patients who identified as gay or bisexual men and were either using PrEP, had declined the offer of PrEP, had stopped PrEP, or had been assessed as ineligible for PrEP (n = 39 of whom n = 5 (13%) identified as trans, median age 31 years and interquartile range 14 years), healthcare professionals involved in PrEP provision (n = 54 including specialist sexual health doctors and nurses of various grades, PrEP prescribing general practitioners, health promotion officers, midwifes, and a PrEP clinical secretary), and clients (n = 9) and staff (n = 15) of non-governmental organisations with an HIV prevention remit across Scotland. We used thematic analysis to map key barriers and facilitators to priority areas that could enhance

participants were assured that raw data would remain private and confidential and would not be shared beyond the use of anonymized illustrative quotes in publications about the research. In line with the ethical approvals, participants did not consent to sharing of their full transcripts. Data access requests may be made to the Glasgow Caledonian University Research Ethics Committee (hlsethics@gcu.ac.uk) and the South East Scotland National Health Service Research Ethics Committee (Sandra.Wyllie@nhslothian.scot.nhs.uk).

**Funding:** This work presents independent research funded by the Scottish Government Chief Scientist Office (reference number HIPS/17/47 awarded to CSE, PF, JF, LM, IY, JS, RN, DC, and NS) – https://www.cso.scot.nhs.uk/. Sponsor's website https://www.gcu.ac.uk/. LM was funded by the UK Medical Research Council (https://www.ukri.org/councils/mrc/) and Chief Scientist Office of the Scottish Government Health and Social Care Directorates (https://www.cso.scot.nhs.uk/) at the MRC/CSO Social & Public Health Sciences Unit, University of Glasgow (MC_UU_12017/11, SPHSU11; MC_UU_00022/3, SPHSU18). The funders had no role in study design, data collection and analysis, decision to publish, or preparation of the manuscript.

**Competing interests:** I have read the journal's policy and the authors of this manuscript have the following competing interests: CSE reports research grants from National Institute of Health Research UK, Chief Scientist Office of Scotland, Engineering and Physical Sciences Research Council, UK Clinical Research Collaboration, Health Protection Scotland, and European Centres for Disease Control. PF reports research grants from National Institute of Health Research UK, Australian Research Council, and Chief Scientist Office of Scotland. RN reports research grants from National Institute of Health Research UK and Chief Scientist Office of Scotland and non-executive director membership of the Board of Public Health Scotland from April 2020. DC has provided expert advice on projects unrelated to prescribing to Gilead, manufacturer of Truvada and Descovy HIV PrEP. Payment was made to the charity British HIV Association (BHIVA) and no direct payment or benefit was received. LM reports research grants from National Institute of Health Research UK, Australian National Health and Medical Research Council, and Chief Scientist Office of Scotland.

adherence and retention in care. We used implementation science analytic tools (Theoretical Domains Framework, Intervention Functions, Behaviour Change Technique Taxonomy, APEASE criteria) and expert opinion to systematically generate recommendations.

## Results

Barriers included perceived complexity of on-demand dosing, tendency for users to stop PrEP before seeking professional support, troublesome side-effects, limited flexibility in the settings/timings/nature of review appointments, PrEP-related stigma and emerging stigmas around not using PrEP. Facilitators included flexible appointment scheduling, reminders, and processes to follow up non-attenders. Examples of the 25 recommendations include: emphasising benefits of PrEP reviews and providing appointments flexibly within individualised PrEP care; using clinic systems to remind/recall PrEP users; supporting PrEP conversations among sexual partners; clear on-demand dosing guidance; encouraging good PrEP citizenship; detailed discussion on managing side-effects and care/coping planning activities.

## Conclusions

PrEP adherence and retention in care is challenging, reducing the effectiveness of PrEP at individual and population levels. We identify and provide solutions to where and how collaborative interventions across public health, clinical, and community practice could address these challenges.

## Introduction

Oral HIV pre-exposure prophylaxis (PrEP, tenofovir disoproxil/emtricitabine) is a highly effective biomedical intervention to reduce HIV acquisition [1, 2], central to the elimination of HIV transmission [3, 4]. Worldwide implementation of PrEP is accelerating but coverage remains patchy [5] and current evidence suggests that adherence to PrEP, critical for efficacy [1, 2], and retention in care are challenging [1, 6–9]. A recent global meta-analysis showed that 38% of PrEP users had suboptimal adherence and 41% had stopped taking PrEP within six months of initiation [9]. Factors associated with poor adherence and PrEP discontinuation may differ according to cultural context and population. However, commonly identified factors among groups at elevated risk for HIV in diverse settings include younger age, being a transgender woman, socio-economic deprivation, lower educational attainment, unemployment, using on-demand dosing, side-effects, PrEP-related stigma, and substance use [9–16]. Cessation of PrEP may happen because of a perceived reduction in HIV acquisition risk [17], which may or may not be accurate.

Despite the burgeoning literature documenting real-world implementation of PrEP across the globe [e.g., 18–24], research drawing on implementation science to specifically enhance PrEP adherence and retention in care is limited. It is unclear how best to identify and support individuals who do not optimally adhere to, or stop, PrEP but remain at, or return to, a risk of HIV acquisition. We need to establish how to encourage adherence to PrEP and retention in care for individuals with ongoing need, and to establish mechanisms through which users can easily restart PrEP as required. Implementation science tools, with their specific focus on

gaining insights to understand and optimise future health service delivery [25], could assist in this endeavour and help unlock the full potential of PrEP [26, 27].

Scotland became one of the first countries worldwide to implement a national PrEP programme [28]. At the time, there were around 4600 people living with HIV attending specialist care in Scotland [29] and 228 people newly diagnosed with HIV each year, half of whom were gay, bisexual, and other men who have sex with men (GBMSM) [30]. From July 2017, PrEP and all associated monitoring were made available as part of broader HIV combination prevention and sexual health care, free at point of access almost exclusively through sexual health clinics, to those at greatest risk of HIV acquisition [31]. Prescribing followed specialist association guidance [32], but services developed their own local models of delivery, largely within existing budgets. These broadly involved: (1) identifying a patient as a PrEP candidate (see [31] for the PrEP eligibility criteria at the time of this study); (2) provision of PrEP information, baseline screening for HIV, other blood borne viruses (BBVs), sexually transmitted infections (STIs), and renal function; (3) prescribing and dispensing PrEP; and (4) regular in person reviews for HIV, BBV, and STI testing, renal monitoring, adherence support, wider sexual health promotion, and PrEP prescribing [32]. Quantitative outcomes from the programme have been reported as part of routine surveillance [31, 33–35] and within a detailed epidemiological study [36].

We conducted a process evaluation of the first two years of Scotland's national PrEP programme. To date, attempts to conceptualise the implementation of PrEP have tended to be broad and descriptive, typically categorising the whole of PrEP care into four or five stages within a continuous linear 'care cascade' [37–40]. Our approach divided the PrEP care cascade into three stages: (1) awareness and access [41]; (2) initiation and uptake [42]; and (3) adherence and retention in care, and then drilled down to focus on the specific steps within each section. Here we consider adherence and retention in care. We defined *adherence* as taking PrEP in line with medical advice / using PrEP appropriately and *retention in care* as attending PrEP review appointments and staying on PrEP during periods of risk.

We addressed the following research questions:

1. Within PrEP care pathways, where should we intervene (priority areas) to improve PrEP adherence and retention in care?

2. What are the barriers and facilitators to implementing the priority areas for PrEP adherence and retention in care?

3. Which evidence-based and theoretically-informed recommendations could improve PrEP adherence and retention in care?

## Materials and methods

Stage 1 is a retrospective qualitative process evaluation within a larger natural experimental design study evaluating PrEP implementation in Scotland (research questions 1 and 2). Stage 2 involves development of a detailed set of recommendations to improve PrEP adherence and retention in care that were derived from stage 1 findings (i.e., evidence-based) and following consultation, using systematic intervention development approaches from implementation science (i.e., theoretically-informed) (research question 3).

### Data collection

**Participants.** We used multi-perspective purposive sampling to understand the implementation of PrEP adherence and retention in care from diverse viewpoints. In total, 117

participants took part in individual semi-structured telephone interviews (n = 71) or in one of 10 group discussions (n = 46) (September 2018-July 2019). The sample comprised: 39 patients; 54 healthcare professionals (HCPs); nine non-governmental organisation (NGO) clients; and 15 NGO staff from across Scotland. All NGOs had an HIV prevention remit and served GBMSM, trans, and/or Black African communities. Group discussions included one type of stakeholder at a time.

Patients were either using PrEP (n = 23, 59%), had declined the offer of PrEP (n = 5, 13%), had stopped PrEP (n = 6, 15%), or had been assessed as ineligible for PrEP (n = 5, 13%). Current and previous PrEP users included those who took PrEP daily (n = 16, 62% current PrEP users; n = 2, 33% previous PrEP user), on-demand (n = 4, 15% PrEP users; n = 1, 17% previous PrEP user), or both ways (n = 6, 23% PrEP users; n = 2, 33% previous PrEP user) (missing data n = 2 PrEP users, n = 1 previous PrEP user). Patients ranged in age from 20–72 years with just over half (n = 21, 54%) between 25–34 years (median age 31 years, interquartile range 14 years). All self-identified as gay or bisexual men, the majority of whom (n = 34, 87%) were cisgender. Almost all were of 'White British' (n = 31, 80%) or 'Other White' (n = 7, 18%) ethnicity. Two thirds reported a university degree as their highest level of education (n = 26, 67%) and the majority were in employment (n = 34, 87%). The patient areas of residence reflected a mix of relative affluence and deprivation although the most (n = 5, 16.7%) and least (n = 3, 10%) deprived quintiles (according to the Scottish Index of Multiple Deprivation (SIMD), which divides areas into five subgroups according to the extent to which an area is "deprived" [43]) were under-represented. Patients predominantly resided in the middle three quintiles (73%) (data missing for 9 participants).

HCPs were all involved in PrEP implementation in a mix of rural (n = 12, 22%), semi-rural/ urban (n = 8, 15%), or urban (n = 34, 63%) settings, largely reflecting the wider Scottish population distribution. They included specialist sexual health doctors (n = 22) and nurses of various grades (n = 23), some with national PrEP roles, PrEP prescribing general practitioners (who prescribed PrEP where there was no sexual health service on their Scottish island; n = 2), health promotion officers (n = 4), midwives (who staffed the sexual health clinic on their Scottish island; n = 2), and a clinical secretary responsible for PrEP-related administration.

NGO clients were all of Black African ethnicity, predominantly cis-gender women, and not using PrEP.

**Recruitment.** HCPs offered patients the opportunity to take part in the study during routine consultations taking place in four of the 14 regional health boards (responsible for the protection and improvement of their population's health) located in urban cities and providing over 80% of PrEP-related care in Scotland [33]. NGO clients who were either engaged with NGOs *and* attending sexual health clinics (classed as patients above) or only engaged with NGO services (classed as NGO clients above) were invited to participate via interactions with NGO staff. We recruited these and other NGO staff and HCPs across all of Scotland's 14 regional health boards by email invitation.

**Procedure.** All participants provided informed verbal or written consent immediately prior to the interviews/group discussions. We collected data with the aid of a topic guide that included open-ended questions designed to explore participants' experiences and perceptions of PrEP adherence and retention in care, rather than questions based on any theoretical concepts anticipated to influence implementation. Where a participant did not have any lived experience of using PrEP to draw on, they were asked to give a hypothetical perspective when answering questions. Where possible within the group discussions, dialogue between participants (rather than between facilitators and participants) was encouraged. All participants talked from their own and others' perspectives. Patients were offered a £30 (~$38USD) shopping voucher as reimbursement for their time.

Data collection was led by JM, with input from experienced qualitative researchers, PF, IY, and JF. Only researchers involved in data collection (JM, PF, IY, and JF) knew the full personal and contact details of participants in order to satisfy sampling criteria and arrange interviews/group discussions. Participants' contact details were kept separately from their personal information and destroyed after study completion. JM, PF, IY, and JF reviewed and discussed early transcripts for quality assurance purposes. All interviews and group discussions were audio recorded, transcribed verbatim, anonymised, and imported into NVivo software for analysis.

## Data analysis

**Stage 1.**  *1. Within PrEP care pathways, where should we intervene (priority areas) to improve PrEP adherence and retention in care?*

Firstly, JM and PF used the Action, Actor, Context, Target, Time behaviour specification framework [44] to conceptualise the sequential actors, actions, settings, and processes (collectively termed 'steps') that constituted PrEP adherence and retention in care (see Table 1). Secondly, we (JM, PF) iteratively created a series of visualisations of the overall, multi-stepped behavioural system of PrEP adherence and retention in care using available UK guidance on best clinical practice in PrEP provision [32] and transcripts of early interviews and group discussions. Thirdly, we (JM, PF) undertook two separate exercises to inform decisions around which steps to focus on, based on their relative importance. The first exercise involved a comprehensive assessment of the breadth and depth of barrier and facilitator data (research question 2) relating to the patient pathway through PrEP adherence and retention in care to identify data 'hotspots' indicative of steps of more importance, and alternatively, data gaps indicative of steps of less importance, from participants' perspectives. The second exercise was a ranking task with input from specialist doctor team members with real-world clinical experience of providing PrEP services in assorted settings (CSE, RN, JS), who considered factors such as amenability to change and likelihood of being enhanced by intervention, to determine the relative importance of each step. This measurement of frequency and ranking, whilst pivotal in shaping our findings (i.e., most important steps retained as priority areas for recommendation development), was more qualitative than quantitative and involved a degree of subjective interpretation.

*2. What are the barriers and facilitators to implementing the priority areas for PrEP adherence and retention in care?*

We (JM, PF) conducted deductive thematic analysis [45] of the qualitative data concerning barriers and facilitators for each priority area. We used the relative frequency of barriers and facilitators to manage the volume of findings and to ensure we focussed only on those that were deemed most important. This stage ended with the identification of the key barriers and facilitators for the priority areas.

**Stage 2.**  *3. Which evidence-based and theoretically-informed recommendations could improve PrEP adherence and retention in care?*

We treated each of the priority areas independently and analysed each separately using a four-step Behaviour Change Wheel (BCW) [46, 47] approach. The BCW is a meta-theoretical framework, developed from a systematic synthesis of multiple prior concepts, constructs, and theories from a range of disciplines and the use of consensus-building among interdisciplinary experts, for use within behavioural change and implementation science research. It encompasses and links to various analytic tools that (1) aid an understanding of the causal mechanisms underpinning a given behaviour(s) (i.e., the Theoretical Domains Framework (TDF)

**Table 1. The different implementation science frameworks and analytic tools used, their discrete purpose, and example applications.**

| Implementation science frameworks and analytic tools | Discrete purpose | Example application |
|---|---|---|
| The Action, Actor, Context, Target, Time (AACTT) behaviour specification framework [44] | A framework that enables detailed specification of the behaviours performed by multiple agents in the implementation of a complex health intervention (i.e., PrEP). | We used the AACTT behaviour specification framework to clarify and map out in detail the specific behaviours of key stakeholders involved in PrEP adherence and retention in care (which we refer to as 'steps' within the overall behavioural system, and then 'priority areas'). E.g., 'PrEP users stop using PrEP'. |
| The Behaviour Change Wheel [46, 47] | An overarching meta-theory that (1) aids an understanding of the causal mechanisms underpinning behaviour and (2) supports the development of theory-based recommendations to improve behaviour. | Examples pertaining to the specific tools inherent within and linked to the BCW approach are noted below. |
| *BCW purpose 1: Aid an understanding of the causal mechanisms underpinning behaviour.* | | |
| The Theoretical Domains Framework (TDF) [48, 49] | A framework of 14 theoretical domains that explains why or why not a behaviour occurs. | We used the TDF to map key barriers and facilitators to the 14 theoretical domains and understand the factors influencing each priority area. E.g., the key barrier 'PrEP users find it difficult to stop using PrEP because of the social acceptability of PrEP and emerging stigmas around *not* using PrEP' mapped to the TDF domain 'Beliefs about consequences'. |
| *BCW purpose 2: Support the development of theory-based recommendations to improve behaviour.* | | |
| Intervention Functions [46, 47] | A framework of nine broad ways to intervene and drive behaviour change. | We used the Intervention Functions to map from the TDF domains pertinent to each key barrier and facilitator to corresponding Intervention Functions. E.g., the key barrier 'PrEP users find it difficult to stop using PrEP because of the social acceptability of PrEP and emerging stigmas around *not* using PrEP' could be addressed by the Intervention Functions 'Education' and 'Persuasion'. |
| The Behaviour Change Techniques Taxonomy (BCTT) v1 [50] | A framework of 93 behaviour change techniques (BCTs) to specify, in granular detail and using a standardised language, potential intervention content. | We used the BCTT v1 to map from the Intervention Functions relevant to each key barrier and facilitator to specific BCTs, which were then operationalised to the PrEP adherence and retention in care context. E.g., the key barrier 'PrEP users find it difficult to stop using PrEP because of the social acceptability of PrEP and emerging stigmas around *not* using PrEP' could be addressed via the BCTs 'Information about health consequences' and 'Framing/reframing'. |
| APEASE criteria [47] | A framework of six criteria–Acceptability, Practicability, Effectiveness, Affordability, Side-effects/safety, and Equity–to consider when assessing the merit of a recommendation. | We used the APEASE criteria to structure detailed discussions about and appraise our "long-list" of initial recommendations. E.g., we removed an initial recommendation to 'use a range of educational methods to enhance PrEP users' understanding of behaviours and situations that carry a higher likelihood of acquiring HIV and facilitate accurate assessments of when they no longer have a need for PrEP' (operationalised BCT 'Information about health consequences') because of potential Side-effects/safety (is it very difficult to assess risk, especially for non-GBMSM PrEP users). |

[48, 49]) and (2) support the development of theory-based recommendations to ultimately improve the target behaviour(s) [46, 47] (i.e., Intervention Functions [46, 47], the Behaviour Change Technique (BCT) Taxonomy (BCTT) v1 [50], and the APEASE criteria [47]). Further details of the four analytic steps and concomitant tools used are provided below and in Table 1. All coding and drafting of recommendations were completed by JM and double-checked for accuracy, validity, and credibility by PF. Any disagreements were discussed until consensus was reached.

Step 1: We began by systematically theorising the key barriers and facilitators for each priority area using the TDF, a meta-theoretical framework of 14 theoretical domains (e.g., 'Skills',

'Social Influences') known to be important in explaining why behaviours do or do not occur across various populations, settings, and health arenas [48, 49]. Each key barrier and facilitator could be coded against multiple TDF domains.

Step 2: We then specified corresponding Intervention Functions, which are nine broad ways of intervening to change behaviour (e.g., 'Training', 'Enablement') relevant to the TDF domains [46, 47], for each key barrier and facilitator. In doing so, we were able to specify, at a high-level, how we could improve the implementation of each priority area.

Step 3: Drawing on the Intervention Functions and working iteratively with the qualitative analysis in stage one, BCTs were chosen from the 93-item BCTT v1 [50] to describe, in granular detail and using a standardised language, potential intervention content (e.g., 'Instruction on how to perform the behaviour', 'Framing/reframing') that may be helpful to address the key barriers and facilitators. We operationalised the selected BCTs to this particular context to specify an initial "long-list" of recommendations that may enhance PrEP adherence and retention in care.

Step 4: Clinical expert team members (CSE, RN, JS) scrutinised, sense-checked, and short-listed the "long-list" of initial recommendations using the APEASE criteria [47], considering Acceptability, Practicability, Effectiveness, Affordability, Side-effects/safety, and Equity, to produce a final set of evidence-based (stage 1 qualitative work) and theoretically-informed (stage 2 analysis) recommendations. This process resulted in the introduction of a small number of new recommendations, in addition to minor amendments to or merging or deleting of existing recommendations.

### Ethical considerations

The Glasgow Caledonian University Research Ethics Committee (HLS/NCH/17/037, HLS/NCH/17/038, HLS/NCH/17/044) and the South East Scotland National Health Service Research Ethics Committee (18/SS/0075, R&D GN18HS368) provided ethical approval.

## Results

### Stage 1

*1. Within PrEP care pathways, where should we intervene (priority areas) to improve PrEP adherence and retention in care?*

We identified 10 priority areas for intervention within the final visualised behavioural system (Fig 1) of a typical PrEP care pathway for adherence (n = 2) and retention in care (n = 8). These priority areas involved two actors (PrEP providers and PrEP users). Six were interactional (1, 4, 5, 6, 8, and 9) and concerned supporting effective PrEP use, assessing ongoing eligibility for PrEP, discussing and addressing wider sexual health issues, communicating the decision to not provide further PrEP, and exploring reasons for wanting to stop/stopping PrEP. Four were more individually oriented (2, 3, 7, and 10) and concerned PrEP users taking PrEP in line with medical advice, attending PrEP reviews, continuing to use PrEP for as long as required, and stopping PrEP safely.

*2. What are the barriers and facilitators to implementing the priority areas for PrEP adherence and retention in care?*

The key barriers and facilitators relating to our priority areas were diverse and multi-levelled, ranging from the macro to the micro, as shown in Table 2. Here we provide a brief narrative overviewing the details in Table 2 for each of the 10 priority areas along with indicative quotations from participants for context.

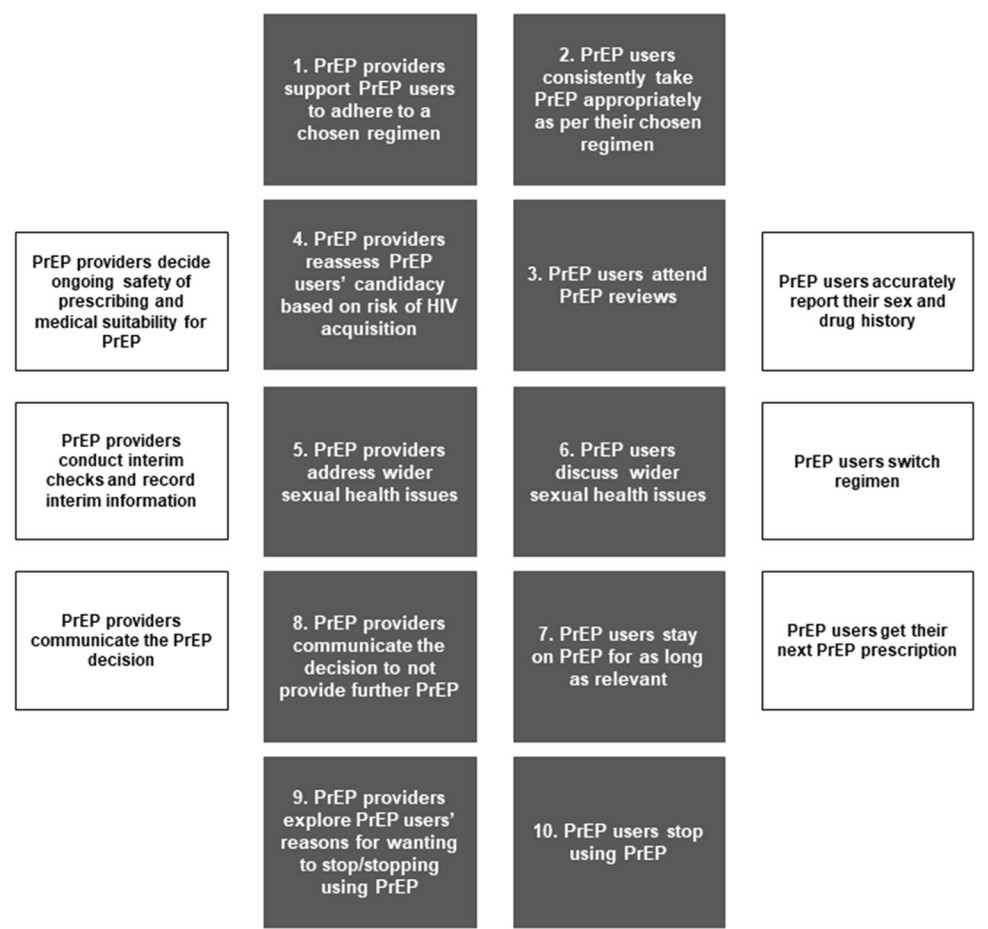

**Fig 1. A schematic of the behavioural system of adherence and retention in care.** White boxes–not selected as a priority area. Grey boxes–selected as a priority area.

Priority area 1. PrEP providers support PrEP users to adhere to a chosen regimen:
Many HCPs were less familiar with, struggled to understand, and found it challenging to make practical suggestions to support correct use of on-demand PrEP. However, clear patient information with example scenarios and visuals aided the provision of accurate dosing advice.

"*I don't know how good I would be if they were saying, "so I'm going to have sex on a Saturday and then I'm going to have sex on a Thursday, when do I actually start and stop it", you know. So, it's case-by-case and I probably still need to refresh my memory a little bit and read up a bit on that. . . most of the people are just taking it every day.*" (HCP)

Priority area 2. PrEP users consistently take PrEP as per their chosen regimen:
Structural issues related to capacity within the sector necessitated PrEP reviews to be implemented through booked appointments (rather than drop-in clinics), which were limited in their availability and created challenges in obtaining the next prescription in a timely manner.

"*The difficulty is where you have DNAs (did not attends) or people just choosing to come to the walk-in clinic for follow-up PrEP and the nursing team not being in a position to be able to do that. . .So, it's. . .trying to fit them in somewhere else and already stretched clinics and*

**Table 2. Key barriers and facilitators to the priority areas for PrEP adherence and retention in care.**

| Priority area | Key barriers | Key facilitators |
|---|---|---|
| *Adherence* | | |
| 1. PrEP providers support PrEP users to adhere to a chosen regimen | • Reliance on user-reported adherence which may over-report good adherence due to a desire to please PrEP providers<br>• Inability to accurately identify when first doses of on-demand PrEP will be needed precludes making practical suggestions to support correct use<br>• Complexity of and unfamiliarity with on-demand dosing, including starting and stopping rules for different scenarios | • Offer practical suggestions to help users remember to take daily PrEP and the 'after' doses when using on-demand PrEP<br>• Provide clear patient information about the various ways to take PrEP with diagrams showing how to take on-demand PrEP |
| 2. PrEP users consistently take PrEP appropriately | • Absence of or disruption to a daily or usual routine (daily users) and inability to predict when sex will occur to trigger first dose for on-demand users<br>• Inflexible clinic appointment processes owing to staff capacity mean PrEP users can run low on or run out of PrEP | • Incorporate taking PrEP into a pre-existing daily routine (if taking PrEP once a day) or a usual routine ahead of planned sex (if using on-demand PrEP)<br>• Receive routine and ad-hoc adherence support from PrEP providers<br>• Put in place reminders to avoid missing a dose<br>• Keep PrEP handy by carrying it or storing it in convenient places |
| *Retention in care* | | |
| 3. PrEP users attend PrEP reviews | • Limited options for where, when, and how to access PrEP reviews<br>• Absence of appointment scheduling, reminder, follow-up and/or other targeted intervention processes<br>• Do not require a new PrEP prescription as using on-demand PrEP or have stopped PrEP in the interim period | • Flexibility in where, when, and how to access PrEP reviews<br>• Appointment scheduling, reminder, follow-up and/or other targeted intervention processes are in place<br>• Value the regular sexual health screening and other health tests and discussions that take place within PrEP reviews<br>• Explicit messaging about the requirement for PrEP reviews at the outset |
| 4. PrEP providers reassess PrEP users' candidacy based on risk of HIV acquisition | • Overlook this aspect of PrEP reviews due to familiarity and routinisation of giving out PrEP and assumptions around ongoing need | • Supporting documents and IT systems prompt this task |
| 5. PrEP providers address wider sexual health issues | • Time constraints of PrEP review appointments | • Generous and/or flexible appointment times for PrEP reviews<br>• Build trusting relationships and familiarity with PrEP users through continuity of care<br>• Trained to deliver brief behaviour change interventions or have the option to signpost PrEP users and/or make direct referrals to other specialist services for appropriate support |
| 6. PrEP users discuss wider sexual health issues | • PrEP reviews feel rushed and are typically only focused on PrEP | • Build a trusting relationship and familiarity with PrEP providers through continuity of care |
| 7. PrEP users stay on PrEP for as long as relevant | • Experience or are concerned about side-effects<br>• Sexual partner(s) is suspicious of PrEP use as they associate it with promiscuity and infidelity<br>• Acquire recurrent sexually transmitted infections while on PrEP | • Positive health, emotional, and social consequences of PrEP |
| 8. PrEP providers communicate the decision to not provide further PrEP | • Inadequate discussion with PrEP users about the risk-benefit of PrEP at the outset owing to a lack of knowledge, skills, and experience by the HCP | • Mention at the start that need for PrEP may change over time and that ongoing eligibility [30] will be assessed and is required to keep issuing PrEP |
| 9. PrEP providers explore PrEP users' reasons for wanting to stop/stopping using PrEP | • PrEP users tend not to discuss their thoughts about stopping PrEP / decision to stop PrEP before stopping | T• here are follow-up and/or other targeted intervention processes in place |
| 10. PrEP users stop using PrEP | • Social acceptability of PrEP and emerging stigmas around *not* using PrEP | • Reduction in self-perceived HIV risk |

*them saying they're running out of medication and then you feeling duty bound to try your best, to try and ensure they don't have gaps in the provision of the medication.*" (HCP)

PrEP users appreciated the adherence support they received from HCPs and reported various strategies to assist them to use PrEP appropriately.

*"When your phone buzzes at 12 o'clock then you know it's time to take your pill."* (PrEP user)

<u>Priority area 3. PrEP users attend PrEP reviews:</u>
Flexibility in where, when, and how to access PrEP reviews and targeted clinic processes to facilitate attendance were key.

*"They can't take the kidney tests in the [outreach] clinic that's dedicated to gay men, because it's in a different venue. . .so, essentially, if at those clinics, if they could take the kidney test as well."* (PrEP user)

Several psychosocial factors were identified, including the importance of managing patient expectations around the requirement for PrEP reviews and the value many PrEP users placed on the regular checks and discussions within PrEP reviews.

*"If you're constantly getting kidney and liver function tests and it comes back positive, then everything's working fine. . .so, that kind of reassures me about my health."* (PrEP user)

<u>Priority area 4. PrEP providers reassess PrEP users' candidacy based on risk of HIV acquisition:</u>
Supportive documents and IT systems were helpful in prompting HCPs to assess continued PrEP eligibility, which could be overlooked.

*"The danger to that is, because you can get a bit complacent about it and think that this is just doing tests and handing out drugs, and not properly reviewing people. . .checking that they still fit the eligibility criteria, and things like that."* (HCP)

<u>Priority area 5. PrEP providers address wider sexual health issues:</u>
Time, continuity of care, and holistic training and/or the ability to signpost or make direct referrals to other specialist services were perceived as critical for HCPs to address wider sexual health issues.

*"These can potentially be quite lengthy and complex dialogues that aren't necessarily going to be able to be accommodated within a short consultation on a three-monthly basis."* (NGO staff)

<u>Priority area 6. PrEP users discuss wider sexual health issues:</u>
The rushed and typically narrow PrEP focus of PrEP reviews were important barriers to PrEP users discussing wider sexual health issues.

*"They don't really say, well, you know, what's your. . .what are you currently up to? Are you seeing anyone or. . .you know, there's no, kind of, counselling service. . .if that's the right term to use. There's no, kind of, how are you in your life and how are you within your sexual health, kind of thing. There's none of that at all."* (PrEP user)

Some PrEP users reported feeling more comfortable discussing wider sexual health issues when there is continuity of care.

*It just feels safer, actually, there's a bond, there's a trust going on there. . .I mean, you should be able to trust a doctor, but for some reason, I find actually speaking to someone that I've known for a while, actually, I feel a lot more comfortable about that."* (PrEP user)

Priority area 7. PrEP users stay on PrEP for as long as relevant:

Side effects and acquisition of recurrent STIs were important considerations, as were the stigmatising beliefs about PrEP of others (e.g., peers, partners) and PrEP users' own beliefs about the perceived positive consequences of PrEP.

"*I expected those kinds of symptoms with dry mouth and the wee bit funny queasiness maybe but in reality, it was a lot more intense and a lot worse than what I anticipated.*" (Stopped using PrEP)

"*I just feel that it gives me reassurance, both in terms of medical reassurance but also psycho-logical reassurance.*" (PrEP user)

Priority area 8. PrEP providers communicate the decision to not provide further PrEP:

Having clear, upfront discussions with patients about the need to continually assess their individual risk-benefit of PrEP was viewed as beneficial in the instance of HCPs being unable to issue a further PrEP prescription.

"*It becomes an issue when there are some reasons maybe not to give PrEP, there are some side-effects, or there's some effect on renal function. And then having to go back and talk about the risk-benefits again. In lots of people, that tends to be not fully discussed properly, it's kind of glossed over.*" (HCP)

Priority area 9. PrEP providers explore PrEP users' reasons for wanting to stop/stopping using PrEP:

Active or opportunistic follow-up and/or other targeted clinical processes are key to engage those who have stopped using PrEP, since they tend not to return to PrEP reviews and discuss their decision with HCPs.

"*Generally, we wouldn't see them again, they just don't access the service, because obviously they feel they don't need it at the moment. So, they don't need PrEP, and they've not been for a sexual health screen. But if they do come back for a sexual health screen, then we'd say, I see you've dropped your PrEP, why was that. And kind of just reflect on it with them, is that the decision that they're happy with, and do they still want to remain off PrEP.*" (HCP)

Priority area 10. PrEP users stop using PrEP:

The increasing social acceptability of and emerging stigmas around *not* using PrEP meant that some PrEP users were hesitant to stop using PrEP.

"*The decision to come off [PrEP] is much harder and more layered than deciding to go on it in the first place. . .with Grindr. . .it's a bit like, well if I'm changing my setting to [HIV] negative instead of being on PrEP, what am I saying? Am I basically saying, one that I'm not valuing my own sexual health and two am I not valuing their sexual health?*" (Stopped using PrEP)

Other PrEP users decided to stop using PrEP due to a reduction in their self-perceived HIV risk.

"*We just got to the point in the relationship where we had a discussion about being exclusive, about sex, about safe sex and made a decision not to see anybody else, be monogamous, and I then took the decision to come off PrEP because I didn't think I needed it anymore.*" (Stopped using PrEP)

### Stage 2

*3. Which evidence-based and theoretically- informed recommendations could improve PrEP adherence and retention in care?*

Our systematic theorisation of the key barriers and analysis, using the TDF [48, 49], led to the generation of an initial 51 recommendations to enhance the implementation of each priority area, specified in both general (Intervention Functions) [46, 47] and highly specific (operationalised BCTs) [50] terms. This "long-list" of recommendations was reduced to 25 final recommendations after applying the APEASE criteria [47] (Table 3 –includes italicised practical suggestions generated by research participants). Full details of our underpinning analyses are provided within S1–S10 Tables.

No recommendations for priority area four (PrEP providers reassess PrEP users' candidacy for PrEP based on risk of HIV acquisition) were retained because recommendations for the other priority areas were deemed more appropriate upon consideration of the APEASE criteria.

## Discussion

### Main findings

We identified 10 priority areas in the PrEP care cascade which could be optimised to improve adherence and retention in care. PrEP users, healthcare professionals involved in PrEP provision, and NGO staff and clients identified multiple barriers and facilitators to effective engagement with these priority areas. Using robust methodology with tools from implementation science, we derived 25 specific recommendations to enhance future PrEP implementation. Recommendations range from those at the "micro-level" within interactions between healthcare professionals and PrEP users, which broadly encompassed tailoring PrEP care to the individual, to higher "macro-level" suggestions for collaboration across agencies and provision of a PrEP in a variety of settings to meet diverse needs.

### Strengths and weaknesses

Little work to date, especially in the UK, has used conceptualisations of the PrEP care cascade as a starting point for systematic and focussed service improvement, whilst explicitly using theory and evidence to enhance PrEP implementation. We directly addressed this gap and focussed on adherence and retention in care, where there is known inequity in outcomes for key vulnerable populations [9]. This large study involved a wide range of clinical and non-clinical stakeholders with varied perspectives and priorities, within a national PrEP programme. Our innovative approach draws directly on participant perspectives, uses the cumulative knowledge embodied within theories of implementation [25, 46, 47], and contributes to implementation science through the use of a shared language and depiction of core concepts (i.e., TDF domains, Intervention Functions, BCTs).

We acknowledge that data were generated from a single country in which PrEP was provided free of charge within sexual health clinics. However, many of the recommendations, such as those which relate to tailoring PrEP support to the individual, flexible appointments, and educational information, are likely to be applicable in most settings in which PrEP is provided, even when PrEP is funded by the individual. We conducted the study in the first two years of the PrEP programme and so findings reflect early stage implementation. Some barriers and facilitators may change as the programme matures, for example, as users and providers become more familiar with on-demand dosing. The participants using PrEP were largely

**Table 3. Final evidence-based and theoretically-informed recommendations to improve PrEP adherence and retention in care.**

| Priority area | Final recommendations |
|---|---|
| *PrEP adherence* | |
| 1. PrEP providers support PrEP users to adhere to their chosen regimen | i. PrEP services should give PrEP providers and NGO staff a list of practical tips for taking PrEP to share with PrEP users. *Strategies for daily PrEP and the 'after' doses of event-based PrEP include*: formulating an 'if-then' plan that links taking PrEP once a day to a specific task (e.g., brushing teeth) which remains constant even in the absence of or disruption to a daily routine; marking PrEP use on a calendar or recording it in a diary; setting reminder alarms and/or using a pill organiser; and keeping PrEP handy by carrying it and/or storing it in convenient places. A strategy for starting on-demand PrEP could be to test different approaches to trigger the initial dose and note which approach is the most successful.<br>ii. PrEP services should use a joined-up, multi-method approach to improve PrEP providers' understanding of on-demand dosing to assist them during consultations. *The following approaches could help*: a range of resources (e.g., national, co-produced PrEP provider pocket guide and patient information, short videos, wall-mounted displays) with clear written instructions and visuals depicting correct usage of on-demand PrEP, including examples of when to start and stop for various scenarios, and a quiz with questions about on-demand dosing as part of PrEP training. |
| 2. PrEP users consistently take PrEP as per their chosen regimen | i. PrEP services should create checklists/proformas, based on formal protocols, to prompt PrEP providers to cover adherence-related issues during PrEP initiation and reviews.<br>ii. PrEP providers should emphasise the importance of adherence to minimise the risks of acquiring HIV and developing antiretroviral resistance and provide verbal, written, and visual instructions regarding medication dosing schedule, starting, stopping, and missed doses.<br>iii. PrEP providers should consider offering PrEP users an explicit exercise in goal setting, coping planning (plans to deal with anticipated barriers to achieving these goals), and review of goals to support adherence to their chosen PrEP regimen.<br>iv. PrEP providers and NGO staff (potentially through the use of peer navigators) should support PrEP users to navigate services and online information for appropriate expert support. *Support could include*: providing clear information on how to get further PrEP prescriptions (i.e. clinic-specific processes, managing expectations—PrEP not an emergency, try and plan appointments in advance as clinics can fill up quickly); ensuring PrEP users know they can return to or call the PrEP service for adherence support and have the option to change regimens; and raising awareness of and directing PrEP users to reputable online sources of adherence support.<br>v. PrEP users should consider a range of strategies, including those outlined in priority area 1, to ensure effective use of PrEP and share those they find beneficial with potential/other PrEP users. |
| *Retention in care* | |
| 3. PrEP users attend PrEP reviews | i. PrEP service planners should consider offering reviews in a range of settings (not solely sexual health clinics). *Each service model should incorporate pathways for non-complex PrEP users and those with additional medical complexity.*<br>ii. PrEP services should ensure individualised PrEP care is provided flexibly to meet diverse needs. *Examples include*: implementing PrEP reviews through drop-in clinics as well as booked appointments (as the programme matures); providing evening and weekend access to suit lifestyles and meet local population needs; ensuring there are options for how to book in for the next review (e.g., online, by phone, in-person), with the appointment system open far enough in advance to enable booking in before leaving the premises; and flexibility to provide extra PrEP supply to accommodate longer periods between reviews, if necessary.<br>iii. PrEP services should use existing or introduce new clinic processes, such as an automated text message (SMS) system (with opt-out option), to remind and follow-up PrEP users about PrEP reviews and to try and reengage non-attenders.<br>iv. PrEP services should consider their patient cohort alongside the available evidence to identify characteristics of people likely to miss appointments or not re-attend for PrEP reviews and develop tailored interventions to be delivered at PrEP initiation to improve retention in care.<br>v. PrEP providers and NGO staff should encourage optimal PrEP use by emphasising the health and emotional benefits of PrEP reviews, such as regular HIV and STI testing, renal monitoring and review of 'how things are going', and the importance of discussing stopping PrEP with a PrEP provider. *Information sources may include co-produced patient information and verbal communication.*<br>vi. PrEP users should commit to engaging with regular PrEP reviews, even if they do not require a new PrEP prescription when the next review is due. |
| 4. PrEP providers reassess PrEP users' candidacy for PrEP based on risk of HIV acquisition | No recommendations relevant to this priority area were retained. |
| 5. PrEP providers address wider sexual health issues<br>6. PrEP users discuss wider sexual health issues | i. PrEP services should ensure flexible provision of individualised PrEP care that meets diverse needs. *For example, explore and provide ways of scheduling appointments with built-in flexibility to respond to long-standing inequalities in health and HIV/PrEP literacy during consultations.*<br>ii. PrEP services and NGOs should enhance and maintain good connections across HIV prevention and care and other specialist services, to facilitate easy reciprocal referrals. *Consider carefully the type of support required and which service is best placed to provide it.*<br>iii. PrEP providers and NGO staff (potentially through the use of peer navigators) should support PrEP users to navigate services and online information for appropriate expert support. *Support could include signposting and/or referring PrEP users to other specialist services across and beyond the HIV prevention and care sector, as necessary.* |

*(Continued)*

**Table 3.** (Continued)

| | |
|---|---|
| 7. PrEP users stay on PrEP for as long as it's relevant | i. PrEP services should provide PrEP providers and NGO staff with a list of management strategies for common side effects that they can share with PrEP users.<br>ii. PrEP providers should spend an adequate proportion of PrEP discussions educating PrEP users about possible side-effects and their typically transient nature and reassure against concerns about longer-term issues and create a personalised PrEP care plan, including information on switching regimens. *Reassurance can be provided by drawing attention to the regular reviews offered to PrEP users.*<br>iii. PrEP providers and NGO staff should consider sexual partners' reactions, views, and perceptions when exploring and probing PrEP users' motivations for wanting to stop or having stopped using PrEP, be cognisant of sexual partner influences on PrEP users' decisions to remain on PrEP, and use their professional judgement to encourage and support PrEP users to have wholistic conversations with their sexual partner(s) about the meaning of PrEP and boundaries of the relationship(s). *Share co-produced example phrases that PrEP users could incorporate into discussions.*<br>iv. PrEP providers and NGO staff (potentially through the use of peer navigators) should support PrEP users to navigate services and online information for appropriate expert support. *Support could include: ensuring PrEP users know they can return to or call the PrEP service to discuss side-effects and have the option to change regimens; and raising awareness of and directing PrEP users to reputable online sources of side-effect management.*<br>v. PrEP information and communications should include specific content on PrEP use within the context of relationships to address PrEP stigma, enable supportive and well-informed discussions among sexual partners, and prevent discontinuation of PrEP where there is an ongoing identified need. *Ensure that materials are co-produced and that communication routes are acceptable to key populations.*<br>vi. PrEP information and communications should include education on the positive health impacts of PrEP, as well as the wider social and emotional benefits and value of PrEP, for communities and individuals. |
| 8. PrEP providers communicate the decision to not provide further PrEP | i. PrEP services should use multi-methods (i.e., a combination of two or more approaches) to develop PrEP providers' knowledge of and skills in explaining instances when stopping PrEP may be in a PrEP user's best interests. *For example, develop and educate PrEP providers on guidance that includes examples of situations where the risk of PrEP outweighs the benefits (e.g., the PrEP user is taking medication for another medical condition that may interact with PrEP and worsen their health [51]), co-produce scripts that address a range of literacy needs for common PrEP risk-benefit scenarios, and provide opportunities to shadow, practice, and receive feedback on communicating decisions to stop PrEP.* |
| 9. PrEP providers explore PrEP users' reasons for wanting to stop / stopping using PrEP | i. PrEP services should assess monitoring and evaluation data to identify 'did not attends' and those overdue a PrEP review and attempt to make contact to discuss decisions to stop using PrEP and reengage them with PrEP care, as appropriate. |
| 10. PrEP users stop using PrEP | i. PrEP and wider sexual health resources and communications should inform of all options for HIV prevention, emphasise the importance of choices, and explain the 'seasons of risk' concept to address emerging stigmas around *not* using PrEP. *Ensure that materials are co-produced and that communication routes are acceptable to key populations.* |

representative of people on PrEP in Scotland at the time (i.e., almost exclusively GBMSM) [31, 33] and, despite our efforts, women and trans and gender diverse people are relatively underrepresented. The lack of diversity among the PrEP using population in Scotland means that the experience and perspectives of healthcare professionals may largely only relate to providing PrEP care to cisgender GBMSM. Thus, our findings lack specificity for and may be limited in their generalisability to other key populations affected by HIV.

## Findings in context of other studies

Our findings build on those from several other studies which have highlighted various barriers to PrEP adherence and retention in care and are in keeping with many of these [7, 14–16, 52]. Furthermore, our recommendations are broadly aligned with elements of recommendations from other authors and public health agencies, for example, co-production of materials [53] and support in navigating healthcare systems (e.g., Prepster [54]). Similarly, embedding PrEP delivery within combination prevention together with a focus on broader sexual wellbeing, inherent within several of our recommendations, was successful in maintaining young men who have sex with men of colour on PrEP in a small feasibility pilot [55]. It is also a model of care recommended within PrEP guidelines [e.g., 56]. The use of text reminders to attend healthcare appointments and adhere to medication has been successfully used in many health areas, including for PrEP, supporting our recommendation to use automated text reminders [57, 58]. However, some promising interventions that could become important steps in this

stage of the PrEP care cascade, for example, the use of peer navigators [59, 60] to improve patient engagement and increase adherence, have not yet been deployed in Scotland hence we have not specified recommendations to enhance their implementation. To our knowledge, no previously published guidance [e.g., 61] has used the rigorous approach to generating recommendations that we took or provided such a comprehensive list of recommendations focussed on improving PrEP adherence and retention in care.

There are examples of effective interventions to improve medication adherence for other disease areas including for people living with HIV taking antiretroviral medication and other conditions requiring long term drug therapy [62–64]. Although these relate to people already diagnosed with a chronic condition which requires long term medication rather than people trying to avoid an infection, there are similarities with our findings. Adaptation of these existing interventions could be useful to improve PrEP adherence and retention in care [65] and vice versa. However, a Cochrane review of improving adherence to and continuation of hormonal contraception, which might better approximate PrEP as it relates to prevention rather than treatment, provided less overlap in findings. For example, intensive counselling and reminders may result in only a slight increase in continuation of hormonal contraception although the effect varied by contraception method [66]. However, to date, interventional studies based on published recommendations, and designed to overcome barriers to improve PrEP adherence and retention specifically, are lacking and robust evaluation of the impact of these approaches is scarce.

## Implications for policy and practice

Many of our recommendations highlight the importance of supporting the individual and understanding their concerns and priorities, together with tailored advice and activities to enhance their understanding of PrEP with discussion of specific strategies to help with ensuring that PrEP is taken appropriately and safely at times of risk, through adherence to a suitable dosing regimen(s). All of these are in keeping with a person-centred approach to care. However, we acknowledge that these activities take time within consultations and services may lack adequate resources to fully provide this level of care as they are currently organised. Within the UK context, sexual health service delivery has changed significantly during the SARS-CoV-2 pandemic with face-to-face appointments being reserved for people who are symptomatic and/or have more complex needs. PrEP services have largely shifted to telephone models [67]. The opportunity to deliver some of our recommendations may be more challenging should services continue with more remote and light-touch models of care, but are no less important. However, this could be an opportunity to commission services through NGOs, including the use of peer navigators. Although the future provision of long-acting PrEP formulations [68] could reduce adherence demands in some respects, there will still be a need for regular review and adherence support. Detailed recommendations to enhance adherence such as these may be even more needed.

Across PrEP services more broadly, healthcare professionals and NGO staff may benefit from training to improve their skills and could usefully learn from each other [42]. NGO staff could play a key role in cultural competency training as well as helping to extend the reach of PrEP to key populations that could benefit, thereby helping to reduce inequalities in provision. In settings where generic medication is available, the costs of providing this support may outstrip drug costs and would need to be appropriately funded in the health care and NGO setting.

## Conclusions

The potential for PrEP to have a major impact on HIV transmission relies on people adhering to it and remaining in active follow up as appropriate to their needs. These recommendations

could directly enhance the quality of PrEP care at an individual patient level, inform the development of interventions to improve adherence and retention in care at programme-level, and ultimately contribute to the global public health priority of elimination of HIV transmission by 2030 [27]. More work is needed with people from a wide range of groups who could benefit from PrEP (i.e., women, trans and non-binary communities, people who inject drugs, migrant communities) to ensure that recommendations and interventions are appropriate to all key groups and to avoid inadvertently widening existing health inequalities. Future work should include robust evaluation of implemented recommendations.

## Supporting information

**S1 Table. Priority area 1—A BCW analysis of 'PrEP providers support PrEP users to adhere to their chosen regimen'.**
(DOCX)

**S2 Table. Priority area 2—A BCW analysis of 'PrEP users consistently take PrEP appropriately'.**
(DOCX)

**S3 Table. Priority area 3—A BCW analysis of 'PrEP users attend PrEP reviews'.**
(DOCX)

**S4 Table. Priority area 4—A BCW analysis of 'PrEP providers reassess PrEP users' candidacy based on risk of HIV acquisition'.**
(DOCX)

**S5 Table. Priority area 5—A BCW analysis of 'PrEP providers address wider sexual health issues'.**
(DOCX)

**S6 Table. Priority area 6—A BCW analysis of 'PrEP users discuss wider sexual health issues'.**
(DOCX)

**S7 Table. Priority area 7—A BCW analysis of 'PrEP users stay on PrEP for as long as relevant'.**
(DOCX)

**S8 Table. Priority area 8—A BCW analysis of 'PrEP providers communicate the decision to not provide further PrEP'.**
(DOCX)

**S9 Table. Priority area 9—A BCW analysis of 'PrEP providers explore PrEP users' reasons for wanting to stop/stopping using PrEP'.**
(DOCX)

**S10 Table. Priority area 10—A BCW analysis of 'PrEP users stop using PrEP'.**
(DOCX)

## Acknowledgments

We are very grateful to the users, patients and staff of sexual health services in all 14 Health Boards, Drs Ruth Holman, Dan Clutterbuck, Maggie Gurney, Nil Banerjee, Pauline McGough, Daniela Brawley, Kirsty Abu-Rajab, Hame Lata, Anne McLellan, Alison Currie, Sharon

Cameron, Hilary MacPherson, Janice Irvine, Graham Leslie, Ciara Cunningham, Maggie Watts. We thank staff and clients of HIV Scotland; Waverley Care (SX Project and African Health Project); THT Scotland; Hwupenyu Health and Wellbeing; and Scottish Trans Alliance. We thank Nathan Sparling and Jacqueline Gray for their contributions to the research process.

## Author Contributions

**Conceptualization:** Jennifer MacDonald, Claudia S. Estcourt, Paul Flowers, Rak Nandwani, Jamie Frankis, Ingrid Young, Dan Clutterbuck, Jenny Dalrymple, Lisa McDaid, Nicola Steedman, John Saunders.

**Data curation:** Jennifer MacDonald.

**Formal analysis:** Jennifer MacDonald, Claudia S. Estcourt, Paul Flowers, Rak Nandwani, Jamie Frankis, Ingrid Young, John Saunders.

**Funding acquisition:** Claudia S. Estcourt, Paul Flowers, Rak Nandwani, Jamie Frankis, Ingrid Young, Dan Clutterbuck, Lisa McDaid, Nicola Steedman, John Saunders.

**Investigation:** Jennifer MacDonald, Paul Flowers, Jamie Frankis, Ingrid Young.

**Methodology:** Jennifer MacDonald, Claudia S. Estcourt, Paul Flowers, Rak Nandwani, Jamie Frankis, Ingrid Young, Dan Clutterbuck, Jenny Dalrymple, Lisa McDaid, Nicola Steedman, John Saunders.

**Project administration:** Jennifer MacDonald, Claudia S. Estcourt, Paul Flowers, Jamie Frankis, Jenny Dalrymple.

**Resources:** Jennifer MacDonald, Claudia S. Estcourt, Paul Flowers, Jamie Frankis, Ingrid Young, Jenny Dalrymple, John Saunders.

**Supervision:** Claudia S. Estcourt, Paul Flowers.

**Validation:** Jennifer MacDonald, Claudia S. Estcourt, Paul Flowers, Rak Nandwani, Jamie Frankis, Ingrid Young, John Saunders.

**Visualization:** Paul Flowers.

**Writing – original draft:** Jennifer MacDonald, Claudia S. Estcourt, Paul Flowers, John Saunders.

**Writing – review & editing:** Rak Nandwani, Jamie Frankis, Ingrid Young, Dan Clutterbuck, Jenny Dalrymple, Lisa McDaid, Nicola Steedman.

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
