## [Decision Letter · Decision Letter 0]

13 Jul 2023

PONE-D-23-12955Improving HIV pre-exposure prophylaxis (PrEP) adherence and retention in care: process evaluation and recommendation development from a nationally implemented PrEP programmePLOS ONE

Dear Dr. Estcourt,

Thank you for submitting your manuscript to PLOS ONE. After careful consideration, we feel that it has merit but does not fully meet PLOS ONE’s publication criteria as it currently stands. Therefore, we invite you to submit a revised version of the manuscript that addresses the points raised during the review process. As you can see below, the reviewers ask for comparably minor changes to the abstract. Please pay particular attention to the comments of reviewer 1 in regard to the already existing body of literature in the field as well as other clarification concerning the discussion of results in relation to the sample and respective transferability. 

We look forward to receiving your revised manuscript.

Kind regards,

Daniel Demant, PhD, MPH, GradCertHEd, BAppSocSc

Academic Editor

PLOS ONE

Journal Requirements:

“I have read the journal's policy and the authors of this manuscript have the following competing interests:

CSE reports research grants from National Institute of Health Research UK, Chief Scientist Office of Scotland, Engineering and Physical Sciences Research Council, UK Clinical Research Collaboration, Health Protection Scotland, and European Centres for Disease Control.

PF reports research grants from National Institute of Health Research UK, Australian Research Council, and Chief Scientist Office of Scotland.

RN reports research grants from National Institute of Health Research UK and Chief Scientist Office of Scotland and non-executive director membership of the Board of Public Health Scotland from April 2020.

DC has provided expert advice on projects unrelated to prescribing to Gilead, manufacturer of Truvada and Descovy HIV PrEP. Payment was made to the charity British HIV Association (BHIVA) and no direct payment or benefit was received.

LM reports research grants from National Institute of Health Research UK, Australian National Health and Medical Research Council, and Chief Scientist Office of Scotland.”

Reviewers' comments:

Reviewer's Responses to Questions

**Comments to the Author**

1. Is the manuscript technically sound, and do the data support the conclusions?

Reviewer #1: Yes

Reviewer #2: Yes

2. Has the statistical analysis been performed appropriately and rigorously? 

Reviewer #1: Yes

Reviewer #2: N/A

3. Have the authors made all data underlying the findings in their manuscript fully available?

Reviewer #1: Yes

Reviewer #2: Yes

4. Is the manuscript presented in an intelligible fashion and written in standard English?

Reviewer #1: Yes

Reviewer #2: Yes

5. Review Comments to the Author

Reviewer #1: This is a national process evaluation of a national PrEP programme in Scotland, followed by recommendations for improved programming of PrEP in the country. There are several areas that could be improved in the manuscript.

1. Abstract: the geographically and demographically diverse could be defined better- what do you mean? National representation or urban/rural or? And demographics? Results in abstract: please can you present sex, gender and median age of the popn and cadre of providers? How did the ineligibles for PrEP contribute data towards the barriers and recommendations?

2. Methods- why did researchers combine those who declined/stopped and ineligible for PrEP in the same group? Seems like these would be very different groups to interview, especially the ineligible for PrEP? Why were so many analytic tools used for this analysis? Was there 1 overarching theory or why so many?

Results:

3. How did the ineligibles for PrEP contribute data towards the barriers and recommendations?

4. How was prep candidate defined in Scotland (page 5 line 85)

5. Please include median age and IQR by group and define the # of participants by health cadre (page 8, lines 145-150)

6. Page 29- multi methods (please define) and when would PrEP risk outweigh benefits (examples?)

Discussion

Do the authors feel that your priority areas and recommendations apply to all PrEP programmes, or just among MSM or TW in Scotland? Please consider the generalizability to other popns and settings given the composition of your qualitative data collection.

7. Not sure you can say “no work to date” (page 30, line 424) as there are several studies in Africa and other settings that use theoretical models to understand the PrEP care cascade. Further below you cite several other studies that you build on.

8. Considering your popn of mostly MSM and TW, are comparisons with contraceptive studies in discussion relevant (line 476)

9. Minor edits- On page 5, line 74, there is a REF without a citation

Reviewer #2: Dear Authors,

I commend you for the intense and thorough work on the project and the resulting paper. Your finds are a worthwhile contribution to understanding PrEP implementation from an implementation science perspective.

I have several comments regarding the manuscript.

1. Since this is a process evaluation, it would be helpful to provide a table of the components of process evaluation measurements and how the data for each measure were obtained, and the key players/support.

2. Regarding the data collection section, why were patients who declined PrEP included in your analysis? Can you explain what helpful contribution these 5 patients added to the findings and recommendations?

3. In the data analysis section, you provided a lot of details and reported using a multiple frameworks as you after different phases of analysis. It might be helpful to readers to create a table with the major frameworks used, purpose of using the framework, and example activity for when the framework was used.

6. PLOS authors have the option to publish the peer review history of their article (what does this mean?). If published, this will include your full peer review and any attached files.

Reviewer #1: No

Reviewer #2: No

---

## [Author Response · Author response to Decision Letter 0]

7 Sep 2023

Dear Dr Demant,

Thank you for the supportive comments on our manuscript and the opportunity to make revisions in line with the reviewer’s comments. We have addressed these point by point below and made changes accordingly in the resubmitted manuscript as requested. We have uploaded tracked and clean versions of the manuscript for your attention.

Here is the additional text for the Competing Interests statement, which explains the ethical restrictions to sharing data publicly: Due to the sensitive nature of the questions asked in this study, participants were assured that raw data would remain private and confidential and would not be shared beyond the use of anonymised illustrative quotes in publications about the research. In line with our ethical approvals, participants did not consent to sharing of their full transcripts. Data access requests may be made to the Glasgow Caledonian University Research Ethics Committee (hlsethics@gcu.ac.uk) and the South East Scotland National Health Service Research Ethics Committee (Sandra.Wyllie@nhslothian.scot.nhs.uk). Please use this same statement to update our Data Availability statement.

We confirm that we are happy with our originally submitted financial disclosure and as such do not require any changes to be made to it. 

Finally, we have updated and checked our in-text referencing and reference list, added captions for our Supporting Information files, and double checked our title page, manuscript, and Figure 1 file (using the PACE tool), all of which appear to meet the PLOS One requirements.

Warm wishes,

Professor Claudia Estcourt

Reviewer: 1

Comments to the Authors

This is a national process evaluation of a national PrEP programme in Scotland, followed by recommendations for improved programming of PrEP in the country. There are several areas that could be improved in the manuscript.

Response: Thank you. Please see specific responses below.

Abstract

1. The geographically and demographically diverse could be defined better - what do you mean? National representation or urban/rural or? And demographics? Please can you present sex, gender and median age of the population and cadre of providers?

Response: Thank you for this comment requesting better definition of our sample within the abstract. We have removed the phrase “geographically and demographically diverse” as we agree that it is rather vague and added details concerning the sex, gender, and median age and IQR of our overall patient sample:

“We conducted semi-structured interviews and focus groups (09/2018-07/2019) with patients who identified as gay or bisexual men and were either using PrEP, had declined the offer of PrEP, had stopped PrEP, or had been assessed as ineligible for PrEP (n=39 of whom n=5 (13%) identified as trans, median age 31 years and interquartile range 14 years) …” (tracked page 2, line 29; clean page 2, line 29)

We have also presented the cadre of providers:

“… healthcare professionals involved in PrEP provision (n= 54 including specialist sexual health doctors and nurses of various grades, PrEP prescribing general practitioners, health promotion officers, midwifes, and a PrEP clinical secretary) …” (tracked page 2, line 33; clean page 2, line 32)

In addition, to match the more detailed descriptions of our patient and healthcare professional samples, we have provided some further information about the remit of the NGOs from which NGO clients and staff were recruited:

“… clients (n=9) and staff (n=15) of non-governmental organisations with an HIV prevention remit…” (tracked page 2, line 36; clean page 2, line 35)

2. How did the ineligibles for PrEP contribute data towards the barriers and recommendations?

Response: A core part of our sampling strategy was to purposively recruit those who faced the greatest barriers to using PrEP (including those who were deemed ineligible and those who declined PrEP when offered it), since they could provide a rich source of data relating to barriers, which may pertain to one or more of the three stages of the PrEP care cascade as conceptualised in our study (awareness and access, initiation and uptake, and adherence and retention in care). We therefore sought to collect data from all participants on the whole PrEP care cascade. While those with no or little experience of using PrEP had more to offer on the initial stage than the latter stages; they were asked to contribute hypothetical perspectives on scenarios relating to PrEP adherence and retention in care instead of drawing on lived experience. 

We have added the following text to the manuscript:

“Where a participant did not have any lived experience of using PrEP to draw on, they were asked to give a hypothetical perspective when answering questions.” (tracked page 9, line 178; clean page 9, line 176)

Methods

3. Why did researchers combine those who declined/stopped and ineligible for PrEP in the same group? Seems like these would be very different groups to interview, especially the ineligible for PrEP?

Response: We would like to clarify that all data, regardless of participant grouping (i.e., whether a patient – those using PrEP, those who had declined the offer of PrEP, those who had stopped PrEP, and those who had been assessed as ineligible for PrEP, healthcare professional, or NGO client or staff), were combined and analysed together using the same analytical framework. We wonder, however, if we inadvertently caused confusion around combining subgroups of patients based on their PrEP status specifically, because of the way we presented our patient mix in the abstract (“…patients who were using or had declined/ stopped/been assessed as ineligible for PrEP.”) and participants section of the main manuscript (“Patients were either using PrEP (n=23, 59%) or had declined (n=5, 13%), stopped (n=6, 15%), or been assessed as ineligible (n=5, 13%) for PrEP.”). To avoid any such uncertainty, we have changed the text in the abstract and participants section, as noted below:

“…patients who … were either using PrEP, had declined the offer of PrEP, had stopped PrEP, or had been assessed as ineligible for PrEP.” (tracked page 2, line 31, clean page 2, line 30)

“Patients were either using PrEP (n=23, 59%), had declined the offer of PrEP (n=5, 13%), had stopped PrEP (n=6, 15%), or had been assessed as ineligible for PrEP (n=5, 13%).” (tracked page 7, line 135; clean page 7, line 133)

4. Please include median age and IQR by group and define the # of participants by health cadre (page 8, lines 145-150)

Response: Thank you for this request. We are reluctant to provide the median age and IQR per patient grouping (i.e., according to their PrEP status) as we feel that doing so would lead to comparisons across the subgroups, which was not the purpose of the study. We have, however, added the median age (31 years) and IQR (14 years) of the overall patient sample (as was requested in the abstract). (tracked page 7, line 141; clean page 7, line 139)

We have defined the number of participants by health cadre. In doing so we noticed an error as we had previously stated ‘a midwife’ when it was in fact two midwives. We have now corrected this error. Please see below:

“They included specialist sexual health doctors (n=22) and nurses of various grades (n=23), some with national PrEP roles, PrEP prescribing general practitioners (who prescribed PrEP where there was no sexual health service on their Scottish island; n=2), health promotion officers (n=4), midwifes (who staffed the sexual health clinic on their Scottish island; n=2), and a clinical secretary responsible for PrEP-related administration.” (tracked page 8, line 154; clean page 8, line 152)

5. Why were so many analytic tools used for this analysis? Was there 1 overarching theory or why so many?

Response: Thank you for the opportunity to provide further information regarding our overarching theoretical framework, the Behaviour Change Wheel (BCW), and its associated implementation science analytic tools. We have added some additional text (tracked page 15, line 232; clean page 15, line 230) describing the high-level, two-fold purpose of the BCW, and clarifying how the Theoretical Domains Framework, Intervention Functions, Behaviour Change Technique Taxonomy v1, and APEASE criteria assist in achieving this purpose. The text is copied below:

“The BCW is a meta-theoretical framework, developed from a systematic synthesis of multiple prior concepts, constructs, and theories from a range of disciplines and the use of consensus-building among interdisciplinary experts, for use within behavioural change and implementation science research. It encompasses and links to various analytic tools that (1) aid an understanding of the causal mechanisms underpinning a given behaviour(s) (i.e., the Theoretical Domains Framework (TDF) [48,49]) and (2) support the development of theory-based recommendations to ultimately improve the target behaviour(s) [46,47] (i.e., Intervention Functions [46,47], the Behaviour Change Technique (BCT) Taxonomy (BCTT) v1 [50], and the APEASE criteria [47]). Further details of the four analytic steps and concomitant tools used are provided below and in Table 1.”

While not specific to the BCW approach, the AACTT behavioural specification framework [44] is designed to enable detailed specification of the behaviours performed by multiple agents in the implementation of a complex health intervention. It can thus inform more focused investigations into the barriers and facilitators to the specific behaviours of key stakeholders and enhance the selection of fit-for-purpose intervention content (i.e. recommendations to change behaviour).

Each of the tools used serves a discrete analytical purpose. As suggested by reviewer 2, in addition to the added text to the manuscript, we have also created a Table outlining the different implementation science frameworks and analytic tools used, their discrete purpose, and example applications from our study, to help readers further understand our methodological approach. Please see page 11, line 217 (tracked) / page 11, line 215 (clean).

Results

6. How did the ineligibles for PrEP contribute data towards the barriers and recommendations?

Response: Please see our response to comment 2, above.

7. How was PrEP candidate defined in Scotland? (page 5 line 85)

Response: We have added the reference for the Implementation of HIV PrEP in Scotland: First Year Report, which details the PrEP eligibility criteria at the time we conducted out study. We have also included the following text, after stating that the first step of PrEP delivery models was identifying a patient as a PrEP candidate, on page 5, line 90 (tracked) / page 5, line 89 (clean): 

“… (see [31] for the PrEP eligibility criteria at the time of this study).”

8. Page 29 - multi methods (please define) and when would PrEP risk outweigh benefits (examples?)

Response: Thank you for the opportunity to clarify our recommendations. We have added a definition after use of the term ‘multi-methods’ within Table 3 (clean and tracked page 34, priority area 8): 

… “(i.e., a combination of two or more approaches)”.

We have also provided an example of a situation where the risk of PrEP outweighs the benefits: 

… “(e.g., the PrEP user is taking medication for another medical condition that may interact with PrEP and worsen their health [51])” – page 34, priority area 8 (clean and tracked).

Discussion

9. Do the authors feel that your priority areas and recommendations apply to all PrEP programmes, or just among MSM or TW in Scotland? Please consider the generalizability to other populations and settings given the composition of your qualitative data collection.

Response: We have reviewed the paragraph where we discussed the generalisability of our recommendations to other populations and settings. Regarding the latter, we consider that many of our recommendations, such as those which relate to tailoring PrEP support to the individual, flexible appointments, and educational information, are likely to be applicable in most settings in which PrEP is provided, even when PrEP is funded by the individual, as stated on page 37, line 462 (Tracked) / page 37, line 459 (clean). We feel that this text is sufficient, therefore, we have not made any changes to it.

In terms of generalisability to other populations, we have already highlighted that “The participants using PrEP were largely representative of people on PrEP in Scotland at the time [31,33] and, despite our efforts, women and trans and gender diverse people are relatively underrepresented. The lack of diversity among the PrEP using population in Scotland means that the experience and perspectives of healthcare professionals may largely only relate to providing PrEP care to cisgender GBMSM.” (tracked page 37, line 468; clean page 37, line 465). However, addressing the reviewer’s request, we have added “i.e., almost exclusively GBMSM” following “largely representative of people on PrEP in Scotland at the time” (tracked page 37, line 469; clean page 37 line 466) and a more explicit summary statement regarding generalisability to other populations at the end of the two sentences noted above, as detailed below.

“Thus, our findings lack specificity for and may be limited in their generalisability to other key populations affected by HIV.” (tracked page 37, line 473; clean page 37 line 470)

10. Not sure you can say “no work to date” (page 30, line 424) as there are several studies in Africa and other settings that use theoretical models to understand the PrEP care cascade. Further below you cite several other studies that you build on.

Response: Thank you for this helpful comment. We have amended our sentence to say ‘little work to date, especially in the UK, …’ (tracked page 36, line 449; clean page 36, line 447).

11. Considering your population of mostly MSM and TW, are comparisons with contraceptive studies in discussion relevant? (line 476)

Response: Thank you for the opportunity to reflect on our comparisons with contraceptive studies, which we consider relevant despite our study sample. Although it may be difficult to draw direct comparisons since contraceptive studies involve almost exclusively female participants;

we are not aware of gender differences in the barriers and facilitators to adherence to and continuation of preventative medication. We have therefore not made any changes to this text.

12. Minor edits - On page 5, line 74, there is a REF without a citation.

Response: Thank you for pointing out this omission. We have now added the reference below to page 5, line 79 (tracked) / page 5, line 78 (clean), updated the in-text reference numbering throughout, and amended the reference list accordingly.

Nilsen P. Making sense of implementation theories, models and frameworks. Implement Sci. 2015; 10:53. doi: 10.1186/s13012-015-0242-0.

Reviewer: 2

Comments to the Authors

I commend you for the intense and thorough work on the project and the resulting paper. Your finds are a worthwhile contribution to understanding PrEP implementation from an implementation science perspective. I have several comments regarding the manuscript.

Response: Thank you. Please see specific responses below.

13. Since this is a process evaluation, it would be helpful to provide a table of the components of process evaluation measurements and how the data for each measure were obtained, and the key players/support.

Response: There are many types of process evaluations, on which guidance is emerging (e.g., McGill et al., 2020; Moore et al., 2015). Rather than undertaking a typical process evaluation using programme theory (Moore et al., 2015), our emphasis was on exploring lessons learned from national PrEP implementation and then further developing potential ways of future optimisation. In this way, we have focussed more on drawing on implementation science frameworks and analytic tools rather than on measuring more typical foci (e.g., CMO configurations (Pawson & Tilley, 1997) or traditional measurements of acceptability, fidelity, relative support for key mechanisms of action and unintended consequences). We have now included a Table (tracked page 11, line 217 / clean page 11, line 215) that that outlines the implementation science frameworks and analytic tools used, their discrete purpose, and example applications, as you helpfully suggested in comment 15.

McGill E, Marks D, Er V, et al. Qualitative process evaluation from a complex systems perspective: A systematic review and framework for public health evaluators. PLOS Med. 2020;17:e1003368. doi:

10.1371/journal.pmed.1003368.

Moore GF, Audrey S, Barker M, et al. Process evaluation of complex interventions: Medical Research Council guidance. BMJ. 2015;350:h1258. doi: 10.1136/bmj.h1258.

Pawson R, Tilley N. Realistic Evaluation. Sage, 1997.

14. Regarding the data collection section, why were patients who declined PrEP included in your analysis? Can you explain what helpful contribution these 5 patients added to the findings and recommendations?

Response: A core part of our sampling strategy was to purposively recruit those who faced the greatest barriers to using PrEP (including those who were deemed ineligible and those who declined PrEP when offered it), since they could provide a rich source of data relating to barriers, which may pertain to one or more of the three stages of the PrEP care cascade as conceptualised in our study (awareness and access, initiation and uptake, and adherence and retention in care). We therefore sought to collect data from all participants on the whole PrEP care cascade. While those with no or little experience of using PrEP necessarily had more to offer on the initial stage than the latter stages; they could contribute a hypothetical perspective on scenarios relating to PrEP adherence and retention in care instead of drawing on lived experience, if they understood them. In addition, as sexual behaviour was one of the eligibility criteria, and this is not static, someone who was ineligible on the day of attendance could later become eligible if their behaviour changed. 

We have added the following text to the manuscript:

“Where a participant did not have any lived experience of using PrEP to draw on, they were asked to give a hypothetical perspective when answering questions.” (tracked page 9, line 178; clean page 9, line 176)

15. In the data analysis section, you provided a lot of details and reported using multiple frameworks at different phases of analysis. It might be helpful to readers to create a table with the major frameworks used, purpose of using the framework, and example activity for when the framework was used.

Response: Thank you for this helpful suggestion. As stated in our response to comment 13, we have now provided a Table that outlines the implementation science frameworks and analytic tools used, their discrete purpose, and example applications from our study (tracked page 11, line 217 / clean page 11, line 215).

---

## [Editor Report · Decision Letter 1]

18 Sep 2023

Improving HIV pre-exposure prophylaxis (PrEP) adherence and retention in care: process evaluation and recommendation development from a nationally implemented PrEP programme

PONE-D-23-12955R1

Dear Dr. Estcourt,

We’re pleased to inform you that your manuscript has been judged scientifically suitable for publication and will be formally accepted for publication once it meets all outstanding technical requirements.

Kind regards,

Daniel Demant, PhD, MPH, GradCertHEd, BAppSocSc

Academic Editor

PLOS ONE

Additional Editor Comments (optional):

The authors have addressed all reviewer comments sufficiently.
---

## [Editor Report · Acceptance letter]

28 Sep 2023

PONE-D-23-12955R1 

Improving HIV pre-exposure prophylaxis (PrEP) adherence and retention in care: process evaluation and recommendation development from a nationally implemented PrEP programme. 

Dear Dr. Estcourt:

I'm pleased to inform you that your manuscript has been deemed suitable for publication in PLOS ONE. Congratulations! Your manuscript is now with our production department. 

Kind regards, 

on behalf of

Dr. Daniel Demant 

Academic Editor

PLOS ONE